# OpenReview forum: "SPIN: Self-Supervised Prompt INjection"
_ICLR.cc/2025/Conference — Submitted to ICLR 2025_

### Official Review · Reviewer_mdxK · 2024-11-02

**Soundness:** 2
**Presentation:** 3
**Contribution:** 2
**Rating:** 5
**Confidence:** 4

**Summary:**

This paper introduces SPIN, a new defense against various LLM attacks. SPIN detects these various attacks by constructing self-supervised tasks and reverses them by adding an optimized prefix to the inputs. Experimental results demonstrate that SPIN effectively defends against different types of LLM attacks.

**Strengths:**

1. The proposed algorithm is straightforward and easy to implement, and the paper provides a clear description of the algorithm.

2. The experiments tested various attacks and evaluated results on different datasets.

3. An adaptive attack is introduced, along with evaluation and analysis.

**Weaknesses:**

1. The paper needs to clarify which specific type of attack it defends against, such as jailbreak, prompt injection, or both.

2. The paper should include additional experiments to show FPR and TPR, especially FPR on benign data. Furthermore, it should evaluate on more recent models.

3. The computational cost of the proposed method may be impractical.

**Questions:**

1. I believe the paper should clarify the specific type of attacks it defends against. Generally, prompt injection and jailbreak attacks are distinct. If SPIN is designed specifically for jailbreak attacks, it should state this clearly rather than introducing things like "detect and reverse these attacks (various adversarial and jailbreaks)" in Abstract or “Figure 1 shows our prompt injection defending an adversarial prompt injection.” If SPIN can also defend against prompt injection, it should be compared with more defenses, such as Known Answer Detection [1] and StruQ [2].

[1] Formalizing and Benchmarking Prompt Injection Attacks and Defenses

[2] StruQ: Defending Against Prompt Injection with Structured Queries


2. I recommend including FPR and TPR (ROC curves) in the evaluation. Current evaluation results only contain ASR. Based on the results in Figure 4, the interject task shows a relatively high FPR (over 0.2 at the best threshold). Therefore, metrics like FPR should also be presented in the evaluation to assess whether SPIN results in significant utility loss.

3. The computational cost of the proposed method may be impractical. First, both the repeat and interjection techniques require an additional query, which can be a substantial cost for high-traffic LLM applications (e.g., ChatGPT). Additionally, while the paper states that "detection will remove a large portion of attacks," Figure 7 shows a noticeable gap in ASR with and without reversal, especially on less robust models (e.g., Vicuna). In such cases, the computational overhead of reversal becomes excessive.

4. The evaluation is somewhat lacking. The experiments were conducted only on Vicuna and LLaMA2. I suggest including more models, such as LLaMA3, Mistral, etc., for broader evaluation. Additionally, since the reversal modifies the prompt structure, I would like to see tests on more datasets to assess the extent of utility loss caused by the reversal rather than just using TriviaQA.

Typo: Figure 4 shows "Best Threshold = 5.73" for interject tasks, but the caption states it as 6.55.

---

> ### Author Response · Authors · 2024-11-28
>
> Thank you for taking the time to read through our paper and giving feedback. We are glad that the reviewer finds our paper to be novel and effective. We address the questions and suggestions below:
>
> 1. **Types of Attacks.** Our defense is able to cover both jailbreak and prompt injection attacks. We revised our paper to follow your suggestions within the abstract and figure 1, and include the defenses mentioned within the related work. Below we also include a simple comparison to StruQ [1]:
>
> | Llama 2 | GCG Attack | Ignore |
> | --- | --- | --- |
> | StruQ | 58% | 0% |
> | SPIN (ours) | 1% | 0% |
>
> 1. **TPR and FPR.** Thank you for mentioning the FPR and TPR curves, we have added a discussion of them in the results section. In the interject task, the FPR is calculated by assuming malicious requests without attacks as positive. In actual experimentation, the alignment will actually prevent the malicious requests without attacks from generating the ‘Paris’ token, leading to a lower FPR than shown in the ROC curves.
> 2. **Computational Costs.** Thank you for raising questions about reasoning costs. We agree that our method introduces an extra reasoning cost to the conversation. However, for critical applications where robustness is important, it is not desirable to make the wrong prediction just to make it faster, and our framework can provide safety. Since our latency is in seconds, our algorithm is still realistic to implement, as Interjection and Repeat add less than 2 seconds to the response. In addition, since our method is easily toggle-able, we will only run the defense algorithm when the task is important, and we can choose how many steps to update at inference time, which can significantly accelerate our method.
> 3. **Additional Models and Datasets.**  Thank you for your suggestion on including more model experiments and datasets. Below we demonstrate our defense method on 2 additional models, and our model is still able to be effective. We also showcase results on 2 additional benign datasets to show that our defense method is still able to maintain performance on normal use cases.
>
>
>     Performance on Additional Models with 100 samples
>
>     | **Model** | **JailbreakChat** | **ICA** | **Adv. Instr** |
>     | --- | --- | --- | --- |
>     | Llama 3.1 (No Defense) | 70% | 19% | 60% |
>     | Llama 3.1 + SPIN | 1% | 1% | 0% |
>     | Mistral-7B (No defense) | 100% | 74% | 92% |
>     | Mistral-7B + SPIN | 0% | 2% | 0% |
>
> Performance on Benign datasets with 100 samples (Results are based on actual token generated, rather than logits)
>
> | **Llama-2 7B** | **Hellaswag** | **PIQA** |
> | --- | --- | --- |
> | Base | 44% | 63% |
> | SPIN (Ours) | 44% | 63% |
>
> [1] StruQ: Defending Against Prompt Injection with Structured Queries

---

> > ### Comment · Reviewer_mdxK · 2024-12-02
> >
> > Thank you for the additional experiments, which addressed some of my concerns. However, I still find it difficult to accept doubling the cost for each query. Therefore, I will maintain my score.

---

### Official Review · Reviewer_3oyT · 2024-11-04

**Soundness:** 3
**Presentation:** 3
**Contribution:** 3
**Rating:** 6
**Confidence:** 4

**Summary:**

The paper introduces a defense mechanism designed to detect and counteract jailbreak prompts. The proposed approach, Self-supervised Prompt Injection (SPIN), employs adaptive prompt injection at inference time, allowing it to detect and mitigate adversarial prompts by leveraging self-supervised tasks. By injecting defense prompts, SPIN effectively reduces the success rate of attacks while preserving performance on benign inputs. The paper demonstrates that SPIN achieves a significant reduction in attack success rates, even against adaptive attackers aware of the defense mechanism. Extensive evaluations on benchmarks, including Advbench, indicate the robustness of SPIN, as it shows a notable reduction in attack success rates for both Vicuna and Llama-2 models.

**Strengths:**

The proposed method leverages adaptive prompts, which dynamically align with attacks, rather than requiring training-time adjustments or extensive resources for defense, offering an innovative defense mechanism against unforeseen and adaptive attacks.

**Weaknesses:**

1. The proposed method is built on an argument that "We find jailbreak inputs often leave a trace, such as degrading other capabilities of the LLM in order to achieve a successful attack." However, this claim is not strictly verified. The paper conducts experiments on Llama-2-chat and Vicuna-7b, which, from today's perspective, do not have strong performance compared to models like the Llama 3 series. Thus, if the evaluations and the claim are based on Llama-2-chat and Vicuna-7b, it may hold true that injecting anything—not specifically jailbreaks—will influence the models' capabilities, but not true for other advanced models. The authors should demonstrate the generalization of their argument. This is critical as the proposed method is built on this argument.
2. Another fatal weakness is that the paper mentions predefined thresholds for the repeat and interjection tasks to distinguish between benign and malicious inputs. However, it is unclear how well these thresholds generalize across different models or contexts, and it seems the paper does not discuss how it decides the thresholds either. In practice, the choice of threshold can have a severe impact on the defense performance, as we can infer based on Figure 4.
3. Based on the computational overhead discussed in Section 4.6, the proposed method introduces additional (multiple) LLM inference rounds. Therefore, it should also be compared with other baselines that have similar computational overhead. For example, guardrail methods like the LlamaGuard [1] series have shown strong performance on jailbreak attacks. Thus, the paper should include a comparison with these guardrail defenses.

[1] Inan, Hakan, et al. "Llama guard: Llm-based input-output safeguard for human-ai conversations." arXiv preprint arXiv:2312.06674 (2023) \
[2] https://huggingface.co/meta-llama/Meta-Llama-Guard-2-8B \
[3] https://huggingface.co/meta-llama/Llama-Guard-3-8B

**Questions:**

1. Could the authors elaborate on the process of threshold calibration for the repeat and interjection tasks? Specifically, how robust are these thresholds across different LLMs and configurations?
2. How does SPIN generalize across more LLMs, as mentioned in the weaknesses?
3. How does SPIN perform with prompts or adversarial inputs from specialized domains (e.g., medical, legal)? Are additional defenses required for domain-specific attacks, or does SPIN generalize effectively across diverse content areas?

Update: I have reviewed the authors' feedback and revised my score accordingly.

---

> ### Author Response · Authors · 2024-11-28
>
> Thank you for taking the time to read through our paper and giving feedback. We agree with the reviewer that our method is able to adaptively defend against malicious attacks without the upfront training time costs. Below, we address the questions and suggestions:
>
> 1. **LLM Generalization + Llama Guard.** Thank you for the suggestions of additional models. Following your suggestions, we have included experiments for two additional models below. In both cases, our defense method is still effectively in preventing jailbreak attacks. In addition we have also included comparisons to Llama Guard [1] as you suggested.
>
> Performance on Additional Models with 100 samples
>
> | **Model** | **JailbreakChat** | **ICA** | **Adv. Instr** |
> | --- | --- | --- | --- |
> | Llama 3.1 (No Defense) | 70% | 19% | 60% |
> | Llama 3.1 + SPIN | 1% | 1% | 0% |
> | Mistral-7B (No defense) | 100% | 74% | 92% |
> | Mistral-7B + SPIN | 0% | 2% | 0% |
> | Llama Guard 2 | 56% | 59% | 2% |
> 1. **Threshold Calculations.** For our defense, we agree it is valuable to have a method that is able to generalize across different models and datasets. The same threshold seen for the repeat in Figure 4 is used for both models, and all the datasets that we use. It was calculated by finding the loss for each prompt in a benign and malicious attacks database, then automatically finding the best threshold to separate the classes. This is a threshold that still has great performance on models different from the one it was calculated on. As for Interjection, we can also check whether it generates the correct token ‘Paris’ within the first 10 tokens. This allows the detection to work across models in a more flexible way.
> 2. **Domains.** Our framework is general enough to adapt to prompts from specialized domains. We also agree that domain specific prompts can be useful to further improve the domain, and in addition, SPIN allows the user to customize the defense injections to account for this specialization. The interjection task can be changed to instead ask a more domain-specific question, which will test for specialized use cases.
>
> [1] Hakan Inan, Kartikeya Upasani, Jianfeng Chi, Rashi Rungta, Krithika Iyer, Yuning Mao, Michael
> Tontchev, Qing Hu, Brian Fuller, Davide Testuggine, and Madian Khabsa. Llama guard: Llm-
> based input-output safeguard for human-ai conversations, 2023.

---

> > ### Comment · Reviewer_3oyT · 2024-11-28
> >
> > Thank you for your responses. I’ve updated my score to 5, as the concerns regarding performance on more advanced models have been addressed. However, I’m still unclear about how you determine the appropriate threshold and the reasoning behind its generalizability. Can  you provide more detailed explaination on this aspects? Thanks.

---

> > > ### Author Response · Authors · 2024-12-02
> > >
> > > Thank you for your question and interest. We treat benign prompts from TriviaQA as (0) and malicious attacks with GCG as (1) and we choose the threshold that maximizes the classification accuracy. Empirically, this threshold generalizes well across attacks like AutoDan, ICA, etc., that are not in the thresholding dataset.

---

> > > > ### Comment · Reviewer_3oyT · 2024-12-02
> > > >
> > > > Thank you for your feedback.
> > > >
> > > > From my perspective, the discussion on setting thresholds and evaluating their generalization should be included and emphasized in the main paper. The paper introduces an interesting phenomenon and demonstrates effectiveness in defending against jailbreak attacks, which is commendable. However, the generalization of the threshold is not clearly addressed. A thorough evaluation is needed in this regard, covering unseen benign and malicious data. I think a promising approach to setting the threshold could involve using an ensemble of benign and malicious data.
> > > >
> > > > Currently, I cannot change my score as I need more time to review the comments from other reviewers before making my final recommendation. My only concern at this point is the generalization of the threshold (but it is also one of the most important aspects of a defense method I think). With the discussion deadline approaching, feel free to provide any further clarification on this aspect if you think it’s necessary. I appreciate your patient rebuttal.

---

> ### Author Response · Authors · 2024-12-03
>
> Thank you for all the helpful feedback, we will emphasize the discussion on thresholds and generalization in the revisions. Following your suggestion for an ensemble of benign and malicious data, below is a table of results across different benign and malicious datasets and we show that the thresholding is consistent.
>
> Thresholding on ensemble datasets for Llama-2
>
> | **Benign vs Malicious** | **Theshold** | TPR | FPR |
> | --- | --- | --- | --- |
> | TriviaQA vs GCG | 0.90 | 1.0 | 0.02 |
> | PIQA vs Jailbreak | 0.86` | 0.97 | 0.06 |
> | BoolQ vs ICA | 0.91 | 1.0 | 0.02 |
> | TriviaQA vs Jailbreak | 0.87 | 0.927 | 0.028 |
> | TriviaQA vs ICA | 0.92 | 1.0 | 0.025 |
> | BoolQ vs GCG | 0.91 | 1.0 | 0.02 |
> | Combined | **0.895** |  |  |
>
> Accuracy using Different Thresholds:
>
> | Dataset | **Threshold = 0.8** | **Threshold = 0.89** | **Threshold = 1** |
> | --- | --- | --- | --- |
> | TriviaQA | 96% | **97%** | 98% |
> | PIQA | 90% | **94%** | 96% |
> | BoolQ | 98% | **98%** | 98% |
> | GCG | 100% | **100%** | 100% |
> | Jailbreak | 100% | **84%** | 80% |
> | ICA | 100% | **100%** | 100% |
>
> Levenshtein Distance Distribution for various datasets
>
> |  | Min | 25th Percentile | Mean | Median | 75th Percentile | Max |
> | --- | --- | --- | --- | --- | --- | --- |
> | TriviaQA | 0.015 | 0.317 | 0.415 | 0.386 | 0.488 | 1.229 |
> | PIQA | 0.095 | 0.170 | 0.315 | 0.245 | 0.297 | 1.852 |
> | BoolQ | 0.274 | 0.345 | 0.407 | 0.396 | 0.457 | 1.369 |
> | Malicious + GCG | 1.01 | 1.176 | 1.224 | 1.222 | 1.273 | 1.452 |
> | Malicious + Jailbreak | 0.846 | 1.287 | 1.352 | 1.331 | 1.605 | 1.681 |
> | Malicious + ICA | 1.662 | 1.672 | 1.681 | 1.682 | 1.688 | 1.703 |

---

### Official Review · Reviewer_CK3i · 2024-11-04

**Soundness:** 4
**Presentation:** 2
**Contribution:** 3
**Rating:** 6
**Confidence:** 4

**Summary:**

This paper detects jailbreakings by task redirection. It introduces an additional inference by prompting the model to do something else (repeat the input, output the capital of France, etc). The detection works because (1) the model tends to do the new task if there are no jailbreakings (2) the model tends to reject to respond when there is a jailbreak. Experiments show an improved security against jailbreakings, and a flexible choice of the three proposed filters. The repeat filter asks the model to repeat the instruction, and the detection loss is the Levenshtein Distance between the original user request (ground truth) and the newly generated sentence. The interjection filter redirects the model to answer “What is the capital of France?” with the loss as the logits for ‘Paris’ as the next token. The reversal layer optimizes a prefix to minimize perplexity, which is much larger when there is a jailbreaking.

**Strengths:**

1. It is novel to detect jailbreakings by prompt injections. The method is well motivated to construct a task where we know the ground truth, and judge using a loss function whether the redirected input leads to the expected response. This seems to be an initial paper that uses prompt injections for good.

2. The authors design three filters and demonstrate their individual defense performance with the computational cost, and show that we could choose different combinations according to the defense case.

3. The experiments are very comprehensive, highlighting the detection advantage. Most notably, it shows good defense performance against adaptive attacks. The authors carefully design adaptive attacks that optimize with the defense, and show that this optimization becomes much harder.

**Weaknesses:**

1. The title is misleading, and it seems to be proposing a new prompt injection attack. However, the paper aims to detect jailbreakings using the idea of prompt injection. Also, I think only the interjection is doing prompt injection as there are two conflicting instructions. Repeat is adding a higher level instruction that treats the original instruction as the data. Reversal is not for detection - it is a prevention defense by optimizing prefixes to have less perplexity. A clearer way to present is necessary.

2. For Table 1, can you explain why repeat and interjection costs less than standard inference? I thought it requires an additional inference, so the overall computation should be more than doubled.

3. The defense assumes the model is vulnerable to prompt injections. What if the model is secure against them? Does it invalidate some proposed defense filters? GPT-4o mini implemented with the instruction hierarchy defense against prompt injections could be a good model to test.

**Questions:**

See weaknesses.

---

> ### Author Response · Authors · 2024-11-28
>
> Thank you for taking the time to read through our paper and giving feedback. We are glad that the reviewer finds our paper to be a novel idea way for prompt injections to enhance the model’s safety. We address the questions and suggestions below:
>
> 1. **Title.** Thank you for bringing up the title. We added to the title ‘D-SPIN: Defensive Self-supervised Prompt INjections’ to clarify that we mean to introduce a defense method.
> 2. **Inference Runtime.** In Table 1, we list out only the cost of doing the check by itself. We will make sure this is more clear in the table. In the case of interjection, since we only look at the initial 10 tokens, we do have to wait as long compared to standard inference. The same is true for repeat because these language models are so verbose that repeating the question is usually shorter than the answering the question.
> 3. **Language Models.** Thank you for your question on different language models. In our paper we tested on various models with Llama-2 already being more secure against prompt injections. Even in the stricter model, because our defense method adds on to the alignment, it only improves on the safety of the base model. We also include a few brief experiments on Llama-3 below to show that a secure model can still be improved by our defense:
>
> Performance on Additional Model with 100 samples
>
> | **Model** | **JailbreakChat** | **ICA** | **Adv. Instr** |
> | --- | --- | --- | --- |
> | Llama 3.1 (No Defense) | 70% | 19% | 60% |
> | Llama 3.1 + SPIN | 1% | 1% | 0% |

---

### Official Review · Reviewer_JLkV · 2024-11-04

**Soundness:** 3
**Presentation:** 2
**Contribution:** 2
**Rating:** 5
**Confidence:** 4

**Summary:**

This paper proposes a method to automatically detect and repair attacker prompts, gaining insights from the relationship between malicious cues and model capabilities. It leverages self-supervision techniques to identify adversarial attacks and implements a inference-time multi-layered defense mechanism against adversarial trigger attacks. The method also aims to complicate the adaptive attacker by considering more constraints.

**Strengths:**

1. The paper presents a novel self-supervised detection scheme, "repeat" and "interjection," based on the impact of jailbreak attacks on LLM capabilities.
2. Building on this detection method and generating prefixes based on perplexity, this work proposes a multi-layer defense mechanism with multiple components. Combining "repeat" or "interjection" with "reversal" to fix malicious prompts appears feasible, achieving a balance between overhead and effectiveness. Additionally, it considers the challenges of adaptive attacks under the defense mechanisms and predicts an increase in the difficulty of future attacks.
3. The paper evaluates a comprehensive range of attack methods and defense mechanisms, providing comparisons with various existing works.

**Weaknesses:**

1. The figures and captions in the paper are inconsistent. For example, in Figure 1, the caption mentions "The blue example," which does not exist in the figure.
2. The overall insight is not prominent enough, the reasons for selecting the "repeat" and "interrupt" tasks are unclear, and there is no other ablation about the prompts of these two tasks.
3. The related work section does not provide a comprehensive introduction to the existing defense mechanisms against jailbreak
4. The number of evaluation models is limited, focusing on only two 7B models.

**Questions:**

1. Why is the defense suffix appended at the beginning? Is there any ablation study to support this choice?
2.The "best thresholds" mentioned in the title (0.89 and 6.55) do not match the values shown in the figure (0.90 and 5.73). Can this inconsistency be clarified?
3. Since the thresholds appear to be fixed, when the dataset is updated, will these thresholds change across different datasets?
4. In real-world scenarios, the "repeat" and "interjection" tasks may affect the outputs required for benign prompts. Is this jailbreak detection mechanism limited in practical use cases?
5. In Figure 5, it seems that the paper by Jain et al. (2023) is not cited. I could not find this reference in the bibliography.

---

> ### Author Response · Authors · 2024-11-28
>
> Thank you for taking the time to read through our paper and giving feedback. We are glad that the reviewer finds our paper to be novel and effective. We address the questions and suggestions below:
>
> 1. **Figures and Captions.** Thank you for catching that typo. We originally meant to refer to the Blue box which represents our defense method and we have made that clear now in our revision.
> 2. **Choice of Detection and Thresholds.** Thank you for your inputs on the insights. We chose the two detection tasks because they had objective ways to measure the correctness, and were easily answerable on benign prompts for most models. Our approach is also general enough that users can design their own questions within the interject task. In addition to our ablation studies in figure 7, below we also include a few ablation on different prompts.
>
> **Choice of Prompts on Vicuna**
>
> | Task | Jailbreak ASR |
> | --- | --- |
> | ‘Repeat’ | 0% |
> | ‘Keep everything the same’ | 0% |
> | ‘Paraphrase the following’ | 5% |
> | ‘Is the following request harmful:” | 60% |
> | ‘What is the Capital of Paris’ | 56% |
> | ‘Who is the 1st President of the US’ | 57% |
> 1. **Related Work on Existing Defenses.** We have made a new paragraph in Related work called Defense Methods which furthers work relating to alignment, and discusses more defense methods, including a few that we compare our own method with.
> 2. **Additional Models.** Thank you for the suggestions about additional model experiments. Following your suggestion, we included 2 new models to demonstrate our defense method’s broader applicability. Our defense method is effective across these different models.
>
> Performance on Additional Models with 100 samples
>
> | **Model** | **JailbreakChat** | **ICA** | **Adv. Instr** |
> | --- | --- | --- | --- |
> | Llama 3.1 (No Defense) | 70% | 19% | 60% |
> | Llama 3.1 + SPIN | 1% | 1% | 0% |
> | Mistral-7B (No defense) | 100% | 74% | 92% |
> | Mistral-7B + SPIN | 0% | 2% | 0% |
> 1. **Defense Suffix.** In our ablation studies we found that appending our tokens at the front or end had similar results, and we chose to append it in the front to clearly differentiate from existing GCG attacks. This is a design choice and will work just as well at the end. Thank you also for catching the threshold typo, and we have revised it in the updated version.
> 2. **Practical Use Case.** We find your question on practical use cases to be important, and since our ‘repeat’ and ‘interjection’ tasks are for detection, the final output generated will be affected on benign prompts. On TriviaQA performance we showed that our defense method is able to maintain similar performance, and below we include two more datasets to show real-world practicality
>
> Performance on Benign datasets with 100 samples (Results are based on token generation, rather than max logits)
>
> | **Llama-2 7B** | **Hellaswag** | **PIQA** |
> | --- | --- | --- |
> | Base | 44% | 63% |
> | SPIN (Ours) | 44% | 63% |
> 1. **Citation.** Thank you for bringing this to our attention, we have now properly added Jain et al’s paper to our bibliography

---

### Meta-Review · Area_Chair_cKEC · 2024-12-15

**Metareview:**

The reviewers and the Area Chair (AC) acknowledge and appreciate the efforts made to improve the paper. However, the reviewers expressed concerns about the proposed technique, particularly the doubling of costs for each query. Additionally, the generalization of the threshold remains a significant issue, and the choice of models used in the evaluation was also questioned. Given these observations, the AC recommends resubmission after addressing all the concerns raised by the reviewers.

**Additional Comments On Reviewer Discussion:**

Reviewer JLkV raised concerns about the evaluation, specifically that it focused only on two 7B models. In their response, the authors included additional evaluation models, but none exceeded 7B in size. These new experiments did not directly address the reviewer’s primary concern.

Reviewer CK3i questioned the assumption that the model is inherently vulnerable to prompt injections and highlighted the possibility that the proposed method could be invalidated if the model is secure against such attacks. The authors' response was not sufficiently convincing to address these concerns for either the reviewer or the Area Chair (AC). A more compelling approach would be to test the proposed techniques on a highly robust model, such as the latest Llama, to evaluate their effectiveness.

---

### Decision · Program_Chairs · 2025-01-22

Reject